# The Prevalence of Arctic Multilayer Clouds and their Observed and Modelled Characteristics

Gabriella Wallentin<sup>1</sup>, Luisa Ickes<sup>2</sup>, Peggy Achtert<sup>3</sup>, Matthias Tesche<sup>3</sup>, and Corinna Hoose<sup>1</sup>

Correspondence: Gabriella Wallentin (gabriella.wallentin@kit.edu)

Abstract. Multilayer clouds (MLCs) are common in the Arctic. With a limited-area setup and 2.5 km horizontal grid spacing, 32 ICON simulations from 22 August to 23 September 2020 were analysed to examine the MLC abundance and characteristics across the Arctic. The model was evaluated against observations from the MOSAiC campaign. An immersion freezing parameterisation was developed to capture the local ice-nucleating particle concentration, increasing the cloud ice number concentration by 14% at temperatures above -12°C. Overall, the model captured most cloudy events with a dry (moist) bias at lower (higher) altitudes. Simulated water paths were underestimated, roughly 3-fold for liquid water and 100-fold for frozen hydrometeors.

A 44%-67% MLC occurrence, smoothly distributed across the Arctic region, was simulated. Modelled MOSAiC occurrence frequencies span 55%- 77%, compared to an observed 46%-69%. While large differences in the total MLC occurrence are found, two-layered systems occur with a systematic frequency of about 22%. The sub-saturated layer between cloud layers is typically < 1 km, indicating a high likelihood of the seeder-feeder mechanism (10%-47%), consistent with observations.

#### 1 Introduction

Multilayer clouds (MLCs), vertically stacked cloud layers, occur with a mean global frequency of about 30% (Wind et al., 2010; Liu et al., 2012; Wang et al., 2016; Subrahmanyam and Kumar, 2017; Matus and L'Ecuyer, 2017; L'Ecuyer et al., 2019; Marchant et al., 2020). In the Arctic, these clouds are found to occur more frequently with occurrence frequencies between 29% - 60% (e.g. Herman and Goody, 1976; Tsay and Jayaweera, 1984; Intrieri et al., 2002; Liu et al., 2012; Vassel et al., 2019; Nomokonova et al., 2019; Vüllers et al., 2021; Silber and Shupe, 2022; Achtert et al., 2025). Meanwhile, due to the lack of vertically resolved satellites above 82°N, the MLC classification from space is limited in the high-Arctic region. Thus, the MLC frequency of occurrence remains difficult to ascertain due to instrument limitations and varying classification approaches.

Despite their relatively large global occurrence, few studies have explored these cloud systems in detail. Their formation mechanisms have been discussed (e.g. Herman and Goody, 1976; Tsay and Jayaweera, 1984; Luo et al., 2008; Morrison et al., 2009; Dürlich et al., 2025) and a consensus towards layering due to advection is emerging. However, this is an ongoing field of research and more studies on this topic are required.

<sup>&</sup>lt;sup>1</sup>Institute for Meteorology and Climate Research Troposphere Research (IMKTRO), Karlsruhe Institute of Technology (KIT), Karlsruhe, Germany

<sup>&</sup>lt;sup>2</sup>Department of Space, Earth and Environment, Chalmers University, Gothenburg, Sweden

<sup>&</sup>lt;sup>3</sup>Leipzig Institute for Meteorology, Leipzig University, Leipzig, Germany

30

From an Eulerian point of view, these cloud systems have been studied through observational studies (e.g. Tsay and Jayaweera, 1984; Turner et al., 2018; Lonardi et al., 2022) as well as through simulations (e.g. Herman and Goody, 1976; Mcinnes and Curry, 1995; Harrington et al., 1999; Luo et al., 2008; Chen et al., 2020; Bulatovic et al., 2023; Wallentin et al., 2025). Idealised large-eddy simulations (LES), initialised with thermodynamical profiles, manage to accurately capture the vertical structure of the cloud system for process studies on microphysical processes (e.g. Bulatovic et al., 2023) while the thermodynamic structure remains a source of error for the layering in realistic models with limited data assimilation (Wallentin et al., 2025).

MLCs may interact through microphysics by the seeder-feeder mechanism (Bergeron, 1965) whereby frozen hydrometeors from an upper cloud layer can fall into and seed a lower cloud. Here, a seed may be referred to as a catalyst for ice formation, akin to ice nucleating particles (INPs), provided they survive the sub-saturated layer between the cloud layers. INPs aid the formation of cloud ice; however, the specific chemical composition and morphology necessary to act as a good INP for heterogeneous ice crystal nucleation is still a topic of research (Kanji et al., 2017; Schmale et al., 2021). Acting as a seed for ice nucleation, water vapour may deposit onto the INP to form ice through the deposition nucleation mechanism, while INPs immersed within liquid droplets may nucleate through immersion freezing (Hoose and Möhler, 2012).

Growth mechanisms of cloud ice into snow and graupel include vapour deposition, aggregation, and riming by raindrops and cloud droplets. The seeding of frozen precipitation (ice crystals, snow, or graupel), may, together with riming and secondary ice production (SIP), initiate glaciation in a supercooled liquid or mixed-phase cloud, through the Wegener-Bergeron-Findeisen (WBF) mechanism (Wegener, 1911; Bergeron, 1928; Findeisen, 1938; Korolev, 2007). This process describes the efficient evaporation of liquid cloud droplets in favour of vapour deposition onto ice particles due to the lower water vapour saturation pressure over ice (below 0°C). The increase in precipitation seen in orographic clouds was first ascribed to seeding (Bergeron, 1965; Roe, 2005) and has since been investigated for both internal (seeding within the same cloud) and external seeding for MLCs over the Swiss Alps (Proske et al., 2021; Dedekind et al., 2024) and in the Arctic (Vassel et al., 2019). Vassel et al. (2019) investigated the occurrence of seeding MLCs on Svalbard, and they found a seeding occurrence of 23%. In studies over the Swiss Alps, Proske et al. (2021) found a seeding frequency of 31% for cirrus clouds overlaying lower clouds, investigated using DARDAR, while Dedekind et al. (2024) found a 10% occurrence frequency of seeding for a modelled case study also over the Swiss Alps.

Radiatively, the different cloud layers in MLC systems influence each other through an increase in downwelling longwave radiation. This impacts the net longwave radiative cooling at the top of a lower cloud layer in the MLC system (Christensen et al., 2013; Adebiyi et al., 2020; Jian et al., 2022; Shupe et al., 2013; Turner et al., 2018; Lonardi et al., 2022; Chen and Cotton, 1987; Luo et al., 2008; Chen et al., 2020). Subsequently, cloud top heights of the lower layer of an MLC system are decreased (Christensen et al., 2013; Jian et al., 2022).

In this study, we set up the ICOsahedral Non-hydrostatic (ICON) model (Zängl et al., 2015) to explore MLCs along the Multidisciplinary Drifting Observatory for the Study of Arctic Climate (MOSAiC) track and the Arctic region during a one-month period in the autumn of 2020 (22 August to 23 September). The ICON model has been used globally and regionally in

the Arctic (Schemann and Ebell, 2020; Kretzschmar et al., 2020; Kiszler et al., 2023, 2024; Wallentin et al., 2025) but has not been systematically evaluated in the high Arctic above the pack ice.

We aim to answer these research questions in particular:

- How does the ICON model with 2-moment microphysics perform in the high Arctic?
- What are the simulated MLC occurrence rates at the MOSAiC site and over the high Arctic?
- Do MLCs differ from single-layer clouds? If so, what are the major characteristics?

The paper is structured as follows. The ICON model, the setup used, and the aerosol constraints in place are presented in Sect. 2. Methods for MLC and seeding detection are presented in Sect. 3. The observational data used for model comparison is introduced in Sect. 4. The results are split into three parts. The first result section, Sect. 5.1 presents the impacts of the newly developed immersion freezing parameterisation. Section 5.2 presents a model evaluation at the MOSAiC site, answering the first research question. Section 5.3 explores the occurrence and properties of observed and modelled MLCs and examines the second and third questions. The final section, Sect. 5.4, investigates the regional occurrence of MLCs and differences compared to the MOSAiC site and addresses the second question from a regional perspective. These results are all discussed in Sect. 6.

## 2 Model setup

85

The ICON model (Zängl et al., 2015), version 2.6.6, is used in a limited-area mode. The simulations are performed on the ICON grid, *R02B10*, with a horizontal grid spacing of 2.5 km. This domain is driven by the ICON Global (13 km) analysis product through the initialisation and boundary conditions. The ICON Global analysis is produced with a 1-moment microphysics scheme and nudged with global as well as local data assimilation from the radiosoundings during MOSAiC. The limited-area mode domain is circular, centred on 90°N with a 2110 km radius reaching 71°N (2398120 grid points). The domain extent is shown with a solid line in Fig. 1b. Vertically, the model top reaches 23 km. 120 vertical levels are placed using terrain-following coordinates with a higher density of levels in the lower atmosphere. At 1 km (4 km), the vertical grid spacing is approximately 50 m (100 m). A 2-moment microphysics scheme (Seifert and Beheng, 2006) is used, with prognostic cloud droplets, cloud ice, rain, snow, graupel, and hail. The fast physics, including cloud microphysics and the saturation adjustment (scheme ensuring water vapour saturation resulting in cloud water condensation), is called with a 20 s frequency, while radiation, described by ecRad (Hogan and Bozzo, 2018), is called every 12 min. Convection, both deep and shallow, is considered explicitly resolved. Turbulence is parameterised using a 2<sup>nd</sup> order turbulence scheme by Raschendorfer (2001). A semi-implicit time integration solver is used.

The 22nd of August 2020 to the 23rd of September 2020 is simulated using 32 consecutive simulations. This particular month was selected due to the large abundance of MLCs, determined by the observational product by Vassel et al. (2019) with updates by Achtert et al. (2025). ICON is initialised at midnight (00 UTC) from the ICON Global analysis. The model then runs for 24 hours with 3-hourly updates at the domain boundaries from ICON Global similar to previous cloud modelling

studies using ICON (Heinze et al., 2017; Kiszler et al., 2023). With a non-stationary observational site, due to the drifting of the ice floe, discrepancies in the spatial dimension can be expected when compared to the model output. To study the clouds at the MOSAiC site, we make use of Meteograms. These are 2-dimensional outputs at the daily averaged location of the ship (ICON uses the closest grid box), to enable model output at a high temporal frequency (1 min), comparable to observations (Kiszler et al., 2023).

#### 2.1 Aerosol constraints

Following previous work on the modelling of Arctic MLCs, we notice the importance of correctly describing the CCN and INP in the model (e.g., Bulatovic et al., 2023; Wallentin et al., 2025). To this effect, we constrain the parameterisations by observations.

#### 2.1.1 Cloud ice formation

105

110

ICON has five pathways for ice formation: two homogeneous and three heterogeneous modes. The three heterogeneous nucleation pathways include immersion freezing of cloud droplets, deposition nucleation, and rain freezing. Immersion freezing is the phase change of a cloud droplet that has an INP immersed within. We use the parameterisation by Hande et al. (2015) (from here on H15), which parameterises INPs based solely on temperature. This parameterisation is active between -36°C and -12°C and has been modified in these simulations, as described below. Deposition nucleation is the vapour accommodation onto an INP, whereby the particle nucleates. We use the deposition nucleation parameterisation also by Hande et al. (2015) which is active between -50°C 

et al., 2019) as well as two datasets from the Zeppelin station on Svalbard, Norway (Wex et al., 2019; Freitas et al., 2023). The INP concentration ( $n_{\text{INP}}$ ) with temperature for the different data sets is shown in Fig. 1a. The various locations for the used data sets are spatially shown in Fig. 1b.

Figure 1a shows measured  $n_{\rm INP}$  together with the current immersion freezing parameterisation in the model (H15). To better represent the Arctic INP distribution, an Arctic-appropriate immersion freezing parameterisation is developed. This parameterisation captures a lower concentration of dust at cold temperatures and the abundance of marine INP species at warmer temperatures. At cold temperatures, T < 253 K, an exponential function is fitted to the data, following the functional form of the H15 parameterisation;

130 
$$n_{\text{INP}} = a \cdot \exp(-b \cdot (T - T_{\text{min}})^c) [L^{-1}]$$
 (1)

with a=2.9, b=0.0815, and c=1.45. T is the atmospheric temperature in Kelvin while  $T_{\rm min}=237.15~{\rm K}$ . To better represent the marine INPs active at higher temperatures, we fit a  $2^{nd}$  order polynomial function at temperatures  $253~{\rm K} \le {\rm T} \le 267.65~{\rm K}$  via

$$n_{\text{INP}} = a \cdot T^2 + b \cdot T + c \left[ L^{-1} \right]$$
(2)

with  $a = 1.41 \cdot 10^{-4}$ ,  $b = -7.589 \cdot 10^{-2}$ , and c = 10.2. This two-part fit is named *Arctic fit* and is shown with a black dashed line in Fig. 1a. For this implementation, deposition nucleation and rain freezing are scaled down by a factor of 0.05 to follow a similar scaling as immersion freezing at cold temperatures. Above 267.65 K, rain freezing is the only active freezing mechanism.

## 2.1.2 Cloud droplet activation

Cloud formation requires the presence of aerosols. Cloud condensation nuclei (CCN) are soluble aerosols capable of cloud droplet formation. In general, the concentration of CCN ( $n_{\rm CCN}$ ) in the Arctic is very low, in the range of 10 to 200 cm<sup>-3</sup> (Mauritsen et al., 2011; Dada et al., 2022; Heutte et al., 2025). CCN species in the Arctic include locally emitted sea salt and sulfate from DMS (Schmale et al., 2021) as well as long-range transport of anthropogenic sulfates (Udisti et al., 2016). During MOSAiC, local emissions contributed 80% to  $n_{\rm CCN}$  in autumn 2020 (Heutte et al., 2025).

The cloud droplet activation is performed using the parameterisation by Hande et al. (2016) (Fig. 2b).  $n_{\rm CCN}$  is parameterised for each pressure level with a decreasing concentration with lower pressure. This scheme is highly dependent on vertical velocity and gives a cloud droplet number concentration ( $N_{\rm d}$ ) at the surface of 200 cm<sup>-3</sup> with vertical velocities of 0.1 ms<sup>-1</sup>. To better represent the Arctic CCN distribution, we compare measured  $n_{\rm CCN}$  from the MOSAiC campaign (Koontz et al., 2020) (cleaned from local pollution using a pollution mask from Beck et al. (2022)) with  $n_{\rm CCN}$  from the MOCCHA campaign (Duplessis et al., 2023; Vüllers et al., 2021) in 2018. The data from the end of August to the end of September is shown in Fig. 2a with a mean  $n_{\rm CCN}$  (for all supersaturations) of 19.5 cm<sup>-3</sup> and 20.0 cm<sup>-3</sup> for the MOCCHA and MOSAiC datasets,

Figure 1. (a) INP concentration in units L<sup>-1</sup> air with temperature from observational datasets (coloured markers) together with the immersion freezing parameterisation by H15 (solid cyan). The newly developed immersion freezing parameterisation is named *Arctic fit* (black dashed line), and further shown is the 1-moment immersion freezing parameterisation by Cooper (1986) (blue dashed-dotted line). An immersion freezing parameterisation, developed for MOSAiC during a case study by Wallentin et al. (2025), is shown with dark green dots. (b) Spatial distribution of the campaign tracks and stations used for the INP parameterisation development: MOSAiC mean position during August-September 2020 (brown circle, Creamean et al. 2022), MOCCHA (pink crosses, Porter et al. 2022), INARCO (grey, Creamean 2019), Utqiavik, Alaska, USA (blue triangle, ARCSPIN campaign, Barry 2023), Alert, Canada (red left-pointing triangle, Wex et al. 2019), Villum, Greenland (orange square, Sze et al. 2023), and Svalbard, Norway (green diamond, Wex et al. 2019; Freitas et al. 2023). All datasets only contain measurements in the time period from August to September. Grey and white contours in (b) represent the levels of 15% sea-ice concentration on the 22nd of August and the 22nd of September, respectively. The domain extent is marked by a solid black line.

respectively. A larger peak can be seen in the MOSAiC dataset during mid-September, coinciding with a larger storm over the ship (Shupe et al., 2022).

To better represent the low  $n_{\rm CCN}$  in the high Arctic, we thus scale the cloud droplet activation scheme. A factor of 0.01 (Fig. 2b), following a similar method as Wallentin et al. (2025), was determined to be the optimal scaling factor for the updrafts reached within the domain to reach a similar order of magnitude of  $n_{\rm CCN}$  as the observations.

**Figure 2.** Cloud condensation nuclei concentration for (a) observations based on two campaigns; MOSAiC 2020 (Koontz et al., 2020) and MOCCHA 2018 (Duplessis et al., 2023) for all supersaturations. The black dotted (solid) line shows the 3-day rolling mean for MOSAiC (MOCCHA). Panel (b) shows the CCN parameterisation in the model (Hande et al., 2016) for vertical velocities appropriate for the Arctic. Solid lines signify the scaled version used here.

# 3 Modelled MLC detection and seeding algorithms

We define a cloud by a cloud mass threshold specified as liquid water content (LWC) plus ice water content (IWC) exceeding  $10^{-9} \text{ kg kg}^{-1}$ . This is in line with the radiation solver's definition of when a cloud interacts with radiation (Rieger, 2019). A mass threshold is otherwise a limiting factor. An evaluation of the mass threshold in terms of changes in cloud layer distribution is thus provided in Sect. 5.3.1. Secondly, a saturation condition is introduced such that a cloud layer requires saturation with respect to liquid water or ice. This is to ensure the exclusion of seeding layers, as these may exceed the mass threshold. Thirdly, layers that are within 100 m of each other will be merged into one cloud. However, cloud layers existing without neighbouring cloud layers may be less than 100 m thick.

The seeding frequency for the model is determined by identifying all cloud tops in the model algorithm outlined above, where the relative humidity over ice or water is at or above 100% and the LWC+IWC exceeds  $10^{-9} \text{ kg kg}^{-1}$ . The model layer in the interstitial layer above the cloud top is defined as a "precipitation layer". If the combined mass of graupel, snow, and cloud ice exceeds a specified seeding mass threshold in this layer (while in a sub-saturated environment), the cloud below is flagged as being seeded. This is further combined with a check of whether the cloud above it, the seeder cloud, has a model layer beneath the cloud base precipitating ice, snow, and graupel above the seeding mass threshold. Therefore, layers above the cloud-tops with sporadically high water contents (such as those resulting from supersaturation fluctuations) can be mostly filtered out. The choice of the seeding mass threshold still presents a limitation and results are presented for high to low mass seeding thresholds of  $10^{-6} \text{ kg kg}^{-1}$ ,  $10^{-7} \text{ kg kg}^{-1}$ ,  $10^{-8} \text{ kg kg}^{-1}$ , and  $10^{-9} \text{ kg kg}^{-1}$ .

200

#### 4 Observational data

The observational data, used for model evaluation in the high Arctic, are collected from the MOSAiC campaign (Shupe et al., 2022) where the ice breaker *RV Polarstern* (Knust, 2017) was moored to an ice floe in the high Arctic during 2019/2020.

## 4.1 Thermodynamic variables

Radiosoundings are used for the vertical profile comparison of temperature and specific humidity. During MOSAiC, these were launched 6-hourly (Maturilli et al., 2022). For a continuous temperature profile, a neural-network product using a combination of two microwave radiometers (HATPRO (Humidity and Temperature Profiler) and MiRAC-P (Microwave Radiometer for Arctic Clouds – Passive) is used (Walbröl et al., 2024). For more details on this product, please refer to Walbröl et al. (2024).

## 4.2 Microphysical retrievals

Cloud variables, including LWC and IWC, are taken from two cloud microphysics retrieval algorithms. CloudNet data (Engelmann et al., 2024) are obtained through the approach by Illingworth et al. (2007) using cloud radars, lidar, microwave radiometer, and radiosondes (Griesche et al., 2024). The ShupeTurner algorithm (Shupe et al., 2015) was specifically developed for microphysics retrievals in the Arctic region and utilises radiosondes, radar, lidar, microwave radiometer, and infrared radiometer to retrieve LWC and IWC. Liquid water path (LWP) is measured using microwave radiometers, while ice water path (IWP) is calculated as the column-integrated IWC. The uncertainty in LWC is 15% to 25% (Frisch, 1998; Griesche et al., 2021) and for LWP, a standard deviation of the mean. One of the main differences between the two retrieval approaches is the temperature dependency in the calculation of IWC. While both use radiosonde profiles, CloudNet calculates IWC with an exponential dependency on temperature (Hogan et al., 2006). The ShupeTurner algorithm employs a power law expression. CloudNet reports an associated uncertainty of + 40 % and - 30 % while Shupe et al. (2015) reports an uncertainty of a factor of two on the retrieved IWC. We obtain a measure of confidence in the retrieved values based on whether or not the results of the two products agree within their uncertainties.

## 195 4.3 Observational MLC detection algorithm

To compare with the model output, we use the observational MLC algorithm developed by Vassel et al. (2019) with updates by Achtert et al. (2025). It is based on radiosondes as the major input data and is thus constrained temporally by the radiosonde launches. The algorithm determines the number of layers present in a radiosonde profile by identifying sub-saturated layers in the calculated relative humidity over ice (from relative humidity over water and temperature). The minimum thickness of a layer is set to 150 m. Radar reflectivity is used as a validation to ensure a cloud layer is present. Fog and low cloud layers are not considered as they fall below the lowest radar range gate. The radiosonde uncertainty includes temperature and relative humidity measurements, which have  $0.3^{\circ}$ C and 4% uncertainty, respectively (Vaisala Radiosonde RS41 Measurement Performance). The uncertainty in the temporal and spatial dimensions, however, due to drifting radiosondes and advected clouds at lower levels, cannot be quantified. Here, we report statistics for MLCs classified using only the radiosoundings and

the combined product, to investigate model and method discrepancies. For more details about the method for detecting SLCs and MLCs during MOSAiC, readers are referred to Achtert et al. (2025).

The seeding occurrence is based on sublimation calculations using Maxwell's growth law with cloud ice in the shape of hexagonal plates. Using a mass-diameter power-law, the mass is determined, which is further used to calculate the fall velocity of the particle. Initial crystal sizes were assumed at 400  $\mu m$ , 200  $\mu m$ , and 100  $\mu m$ . A margin of error is allowed when the seeding fall height ends within the vertical grid spacing of the radar.

To efficiently compare the model data with the observations, model variables are extracted during a 30-minute window, starting from the launch of the radiosonde. This is justified through the method of determining the observed MLC occurrence, as 30 minutes before and after the radiosonde launch are used for the classification (Achtert et al., 2025).

#### 5 Results

210

## 5.1 Impacts of the adjusted CCN and INP parameterisations

We use the simulation of the 31st of August 2020 at 18 UTC, to explore the implications of adapting the INP and CCN parameterisations. The domain is large enough to cover the entire temperature range relevant for ice nucleation. The model is initialised at 00 UTC, and after 18 hours of simulation, appreciable differences, if any, should be quantifiable.

The changes to the CCN parameterisation introduce a negligible difference in  $N_{\rm d}$  within the domain (not shown). We filter for more densely cloudy grid boxes (LWC + IWC >  $10^{-6}~{\rm g~m^{-3}}$  and  $N_{\rm d} > 1~{\rm cm^{-3}}$ ) and find a 0.2% decrease in  $N_{\rm d}$  from a mean value of 852 cm<sup>-3</sup> for H15 to 850 cm<sup>-3</sup> for the new CCN scaling. Mass changes are also negligible, as expected due to moisture limitations (not shown).

To investigate the changes in cloud ice, we filter the data for cloudy grid boxes (LWC + IWC >  $10^{-6}~{\rm g~m^{-3}}$ ) and evaluate all temperatures. This illustrates the dominance of cold-temperature ice nucleation, where the H15 parameterisation yields a larger ice number concentration ( $N_{\rm ice}$ ). A 10% decrease in  $N_{\rm ice}$  is found for the *Arctic fit* owing to the reduction of  $n_{\rm INP}$  for immersion freezing, deposition nucleation, and rain freezing. Mean  $N_{\rm ice}$  reach 4.4  $L^{-1}$  for the *Arctic fit* compared to 4.8  $L^{-1}$  using the H15 parameterisation (not shown).

To isolate the in-cloud effect for the addition of immersion nuclei, we filter for single-layer cloud-tops above 261K, in the temperature range not covered by the H15 parameterisation. Similarly, the impact at cold temperatures (243K  $\leq$  T < 261K, to exclude homogeneous nucleation temperatures) is quantified. The box plots in Fig. 3a show a 14% increase in the mean  $N_{\rm ice}$  for the *Arctic fit* (0.024 L<sup>-1</sup>) at warm temperatures compared to H15. Median values remain higher for the H15 implementation (mean 0.021 L<sup>-1</sup>). The presence of cloud ice at these temperatures using the H15 scheme, where no immersion freezing is active, is due to advection from colder regions and rain freezing. At cold temperatures, Fig. 3b, a 17% decrease in the mean  $N_{\rm ice}$  is found (0.93 L<sup>-1</sup> vs 0.77 L<sup>-1</sup> for H15 and *Arctic fit*, respectively). The decrease in  $N_{\rm ice}$  at cold temperatures is expected due to the reduction in  $n_{\rm INP}$  for the *Arctic fit* throughout this temperature range. Meanwhile, the rather small increase in  $N_{\rm ice}$  at warm temperatures may be partially due to the decreased rate of rain freezing.

Overall, the impact of changing the INP parameterisation on the full domain is small. For completeness, cloud ice mass concentrations are shown in Fig. B1 and, as expected, do not increase with the new parameterisation; rather, a 7% decrease is found at warm temperatures. However, the small changes in  $N_{\rm ice}$  can enable local INP activation at higher temperatures (T > 261K), which better represents the Arctic abundance of warm-activating INPs. Thus, for the simulations performed here, we use the *Arctic fit* parameterisation.

Figure 3. Statistics of  $N_{\rm ice}$  of SLC cloud-tops for the H15 parameterisation and the Arctic fit filtered according to cloud-top temperatures in the range  $-12^{\circ}{\rm C} \le {\rm T} 

245

250

255

260

265

270

275

#### 5.2 Model evaluation at the MOSAiC site

Fig. 4 presents the results of the 32 consecutive simulations and the observations; CloudNet (Engelmann et al., 2024) and the ShupeTurner microphysics retrieval (Shupe, 2023). Generally, the model represents the observed clouds remarkably well. The persistent boundary layer clouds are captured. However, the cloud phase tends to be predominantly liquid in these simulated low clouds compared to the retrieved mixed-phase or ice character, visible primarily in the frozen precipitation falling from the liquid cloud tops of the lower cloud layers. A similar problem was identified in Wallentin et al. (2025). A - 5.5°C isotherm is shown in each panel in Fig. 4, where we can note the high altitude of the isotherm until the 7th of September. The model cannot generate primary ice below this altitude (apart from low rates of rain freezing), and any ice present must be supplied through seeding for those low-level clouds to be mixed-phase. Notably, the microphysical retrievals differ at these temperatures. The CloudNet retrieval lacks liquid clouds as well as cloud ice compared to the ShupeTurner algorithm and the model. Likely, these clouds were detected as rain or drizzle and thus are not present in the LWC product. The uncertainty in retrieved IWC, and especially the vertical distribution of cloud ice at these warm temperatures, may be a limiting factor in this discussion.

Most high clouds and events of frozen precipitation are also modelled well, with a tendency towards over-predicting the presence of both. The ice clouds, while more numerous, tend to have a lower frozen water content in the model. Discrepancies compared to the observations may be due to differences in moisture (see below), aerosol availability and its model representation, and overall difficulties in representing MLCs (Wallentin et al., 2025).

Rising cloud tops of the lower cloud throughout each day can be seen in the model output. This showcases a common problem in NWP models: the very active cloud-top mixing process (Sandu et al., 2013). This may also be attributed to an insufficient subsidence rate in the model that allows for larger vertical motions, weak temperature inversions, or a high entrainment rate. The clouds during MOSAiC have been shown to have a low entrainment rate (Neggers et al., 2025), a property that cannot be tuned for in the model.

Figure 5 shows the column-integrated LWP and frozen water path (FWP) for the model and the two microphysics retrievals. We define an FWP category to more easily compare with observed cloud ice. In the model, this includes cloud ice, snow, and graupel, while in the observation, this is classified as ice. LWP is here the column-integrated LWC and thus differs slightly between the retrievals. Meanwhile, the model underestimates the median LWP by a factor of about 2.5 (2 for CloudNet, 3 for ShupeTurner). In the FWP category, the observational products differ more (factor of 1.5), but a larger discrepancy for the model can be seen. Approximately two orders of magnitude difference is found between the datasets, highlighting the lack of cloud ice seen in Fig. 4c. We perform the non-parametric statistical Mann-Whitney U test (Mann and Whitney, 1947) (Appendix A, Fig. A1a) to explore to what extent the distributions differ. Statistical significance (p 

Figure 4. The long-term simulations performed at the MOSAiC site. Shown here are the ShupeTurner microphysics retrieval (Shupe, 2023) (a), the CloudNet microphysical product (b), and the model output (c) between the 22nd of August and the 23rd of September 2020. Liquid water content is shown in green. A frozen hydrometeor category is defined and is shown in blue with dashed outlines. For the model, this includes mass concentrations for the specific hydrometeors: cloud ice, snow, and graupel, while for the observations, the ice water content implicitly includes all modelled hydrometeors. The model is initiated at 00UTC every day and runs for 24 hours; vertical dashed lines indicate the initialisation of the model each day. Observed (Walbröl et al., 2024) and modelled temperature isotherms are outlined in the respective panels for the upper limit of the immersion freezing parameterisation at - 5.5°C (black).

Next, profiles of temperature and specific humidity are examined. The modelled data shown here, and for the rest of the comparisons with observations, only includes data within 30 minutes of the radiosonde launch to capture a similar state as the observations. Mean temperature profiles (Fig. 6a,b) agree well with a small cold bias in the lowest kilometre (max difference 0.8 K at 750 m). A similar small cold bias has previously been seen for ICON over Svalbard (Kiszler et al., 2023). Mean profiles of specific humidity (Fig. 6c,d), on the other hand, differ between the simulations and the observations with the model

**Figure 5.** Box plots of total liquid water path (LWP) including integrated rain and frozen water path (FWP, including cloud ice, snow, and graupel integrated water paths) for the model and the two observational retrieval products; CloudNet (Engelmann et al., 2024) and ShupeTurner (Shupe, 2023).

simulating lower values close to the surface (below 2 km, maximum difference in the mean:  $0.7~\rm g\,kg^{-1}$ ) and higher values at higher altitudes (above 2 km, max difference  $0.2~\rm g\,kg^{-1}$ ) by the model. Similarly to the temperature discrepancy, the dry bias in the lower atmosphere has been recorded over Svalbard (Kiszler et al., 2023). The moister high altitudes may explain the larger occurrence of high clouds in Fig. 4.

# 5.3 Modelled and observed MLCs

285

# 5.3.1 MLC occurrence during MOSAiC

The MLC occurrence for the MOSAiC site is presented in Fig. 7. Shown here are the differences introduced in the MLC occurrence by variations in the cloud mass threshold discussed in Sect. 3.

As expected, a low cloud mass threshold  $(10^{-9} \text{ kg kg}^{-1}, \text{Fig. 7a})$  gives more cloud layers compared to larger mass thresholds, as thinner cloud layers are included. The MLC occurrence decreases from 77% to 55% when increasing the mass threshold from  $10^{-9} \text{ kg kg}^{-1}$  to  $10^{-6} \text{ kg kg}^{-1}$ . With a larger number of MLCs, either the SLCs or the clear-sky fraction has to decrease.

300

305

**Figure 6.** Height-resolved occurrence rate of temperature (a,b) and specific humidity (QV) (c,d) from soundings (a,c, Maturilli et al. 2022) and model simulations (b,d). Mean profiles are shown as dashed lines in red for the observations and black for the model. In panels b and d, the observed mean profile is included for easier comparison.

We notice that the clear-sky fraction changes less (5% to 14%) than the SLC fraction (18% to 31%). This indicates that more profiles change from MLC to SLC by excluding layers of clouds below the mass threshold, rather than from MLC to clear sky. In the subcategories of the MLC occurrence, the distribution of the number of cloud layers within MLCs reveals a robust distribution of 2-layered clouds. The occurrence of three-layer MLCs also remains relatively constant, while MLCs consisting of four or more layers are less abundant and decrease more rapidly for higher mass thresholds. This may indicate that two-and three-layer structures are relatively robust, while cloud systems with four or more layers contain shallower clouds that are highly sensitive to the choice of detection threshold.

The observed MLC occurrence for the MOSAiC site is presented in the last two bars in Fig. 7 using the algorithm by Vassel et al. (2019) with updates by Achtert et al. (2025). A 69% occurrence of MLCs is identified using only the radiosondes (supersaturated layers) as a detection method (RS). This gives a 21% occurrence of SLCs. The profiles are most commonly two-layered, but similarly to the model, a high occurrence of three layers is found, while four or more layers are found less frequently. The inclusion of the validation method (cloud presence) using radar collocation (RS+Radar) reduces the MLC occurrence drastically. Reasons for the large reduction include a large presence of supersaturated layers without cloud mass present, difficulties distinguishing seeding layers, removal of low clouds and fog layers below the lowest radar gate, or attenuation in the radar, limiting high-cloud occurrence. Further details regarding the features and limitations of the observed cloud-layer detection are provided in Achtert et al. (2025). The MLC occurrence reaches 46% with a large proportion of SLCs

Figure 7. Bar charts of cloud occurrence showing the MLC, SLC, and clear sky (CS) distributions for the model and observations. The first four bars show the statistics for each cloud mass threshold defined in Sect. 3 varying between  $10^{-9}$  and  $10^{-6}$  kg kg<sup>-1</sup> as indicated by the label. The subcategories of MLCs show the occurrence of 2, 3, 4, and more than 4 layers, respectively. The observations, the final two bars, are based on the algorithm by Vassel et al. (2019) and updated by Achtert et al. 2025. "RS" indicates cloud occurrence based on radiosoundings, while the "RS+Radar" category shows the MLC occurrence determined by the observational MLC algorithm based on the radiosoundings and validated by radar data.

(33%) and clear-sky (21%). The total cloud fraction of about 80%, as well as the MLC/SLC partitioning, corresponds to findings by Barrientos-Velasco et al. (2025) during MOSAiC. Interestingly, a common occurrence of 2-layered MLCs is found across the model and observations, irrespective of the smaller total MLC occurrence. The similarities in the representation of the approximately 23% occurrence of two-layered clouds may be considered a robust response, while the uncertainty remains in capturing clouds with more than two layers present.

Comparing the model results for different cloud-water mass thresholds to the radiosounding classification indicates that the latter are best resembled when using a threshold of  $10^{-8} \text{ kg kg}^{-1}$ . Meanwhile, the radar inclusion produces values much lower than the lowest mass threshold used in the model. We may not say with certainty that the observations are not underpredicting MLC occurrence. However, model limitations include the imposed mass threshold that defines a cloud, difficulties modelling the observed clouds due to the abovementioned moisture bias or uncertainties regarding the aerosol availability. For the remaining analysis at MOSAiC, a cloud-water mass threshold of  $10^{-9} \text{kg kg}^{-1}$  is used to allow for the presence of shallow clouds in the model.

330

## 320 5.3.2 Macrophysical characteristics of MLCs

We further investigate macrophysical differences between MLCs and SLCs. Cloud thickness, cloud height, and thickness of the sub-saturated layer in between two cloud layers are evaluated and shown in Fig. 8. The observed data are based on the MLCs identified using radiosoundings only.

**Figure 8.** (a) Cloud thickness for MLCs, *1st Layer MLC*, and SLCs for the model (light blue) and observations (dark blue) and MLC gap thickness ("Gap"). (b) Cloud-top (CT) distribution for the respective cloud systems. The cloud boundaries are determined in the observational MLC algorithm from the relative humidity measurement, and heights are retrieved from the radiosonde data (Maturilli et al., 2022). The model data is based on the MLC algorithm with a mass threshold of  $10^{-9}$  kg kg $^{-1}$ . The first layer (lowest cloud in the MLC system) cloud-tops are also extracted from the dataset (1st Layer CT MLC). Horizontal lines indicate the median.

Interestingly, individual layers in the MLC system have a median cloud thickness (446 m in the model data and 480 m in the observations) that is greater than SLCs (298 m in the model data and 354 m in the observations). The modelled thickness of both SLCs and MLCs has a larger spread compared to the observations. This indicates that thicker clouds are simulated compared to what is observed. However, median thicknesses are smaller in the model than observations. This may be explained by a 12% (25%) occurrence of modelled thin MLC (SLC) layers that are less than 100 m thick. These cloud layers would not be included in the observational algorithm. Li et al. (2011) also found that MLCs tend to be thicker than SLCs (at 80°N) but with values between 2 km and 4 km. The *1st Layer MLC* is the lowest cloud in an MLC system, and is thought to be most affected by radiative impacts from upper layers. However, we find that the median thickness of the *1st Layer MLC* is similar to SLCs (39 m and 22 m difference in model and observations, respectively).

The gap thickness ("Gap" in Fig. 8a), the depth of the subsaturated layer separating two cloud layers in the MLC system, reaches medians of 752 m (667 m) for the model (observations). The majority of modelled and observed cloud gaps are less

365

than 3 km deep. This indicates that the occurrence of a double-layered structure with cirrus clouds overlaying boundary-layer clouds is rare. In fact, only 8% (4%) clouds have a gap thickness of > 5 km in the model (observations). These results are in line with the full year analysis of observed MLCs during MOSAiC (Achtert et al., 2025, , their Figure 8) who find that the gap thickness is generally below 3 km.

We further note that MLCs have higher cloud-tops than SLCs (Fig. 8b), a tendency found in both the observations and the model. The difference between MLC and SLC median cloud-top heights for the model (observations) is 2288 m (2532 m). Interestingly, the median cloud-top heights are always found at higher altitudes in the observations than in the model. This may be due to algorithmic constraints in the observational algorithm outlined above and in Achtert et al. (2025). The SLCs are mostly constrained to the lowest part of the troposphere. Indeed, SLCs are rarely found at cirrus level and only 27% (13%) of the modelled (observed) SLCs have a cloud-top height above 7 km.

Cloud-top height suppression, discussed in Christensen et al. (2013) and Jian et al. (2022), due to the presence of an overlaying cloud layer, can partially be seen in Fig. 8b. The median cloud-top height difference between SLC and *1st Layer MLC* is 190 m in the model. In the observations, the opposite is found, where the *1st Layer MLC* is found 116 m higher. Discrepancies in this reduction in cloud-top height may be due to the skewness in modelled SLC top heights. In the observations, the clouds are less constrained to the lowest kilometre (likely due to the omission of the lowest-level cloud layers as outlined above), and thus the median of the cloud top height is larger. Interestingly, the cloud top height suppression for the model is larger than the difference in thickness, indicating that the altitude of the cloud base of the *1st Layer MLC* is simply shifted downwards instead of a thinning of the cloud layer.

## **5.3.3** Seeding occurrence

We find that about half (46%) of the modelled MLCs have been seeded according to the criterion described in Sect. 3. This specification requires that cloud layers have a combined snow, graupel, and cloud ice mass above  $10^{-9} \text{ kg kg}^{-1}$  (seeding mass threshold) falling into the cloud from a sub-saturated layer above. A large dependency on this threshold is observed, and the seeding frequencies using different seeding mass thresholds are tabulated in Table 1. From the observational algorithm (Achtert et al., 2025), we find that with 400 µm ice crystals, 59% of MLCs seed the layer below or, more specifically, have the potential to do so. The respective (potential) seeding frequencies for ice crystals of sizes 200 µm and 100 µm are tabulated in Table 1. We do not currently possess an evaluation method that can single out which ice crystal size is more likely to occur and seed the lower layer.

## 5.4 MLCs across the Arctic

The MOSAiC site, while an excellent site for model comparison, is only a local area. In this section, we explore whether this site is representative of the whole high Arctic region. The large-scale occurrence of MLCs is based on the analysis performed on the full domain outlined in Fig. 1b, where we exclude land areas. The MLC occurrence in Fig. 9 shows similar but lower values than MOSAiC for the respective cloud-mass thresholds. MLC occurrence decreases from 67% to 44% when increasing the cloud mass threshold from  $10^{-9} \text{ kg kg}^{-1}$  to  $10^{-6} \text{ kg kg}^{-1}$ . Interestingly, the difference in the occurrence frequency is

| Model            |     | Obs    |     |
|------------------|-----|--------|-----|
| $10^{-9}$        | 48% | 400 μm | 59% |
| 10-8             | 27% | 200 μm | 44% |
| 10 <sup>-7</sup> | 17% | 100 µm | 27% |
| $10^{-6}$        | 9%  | _      |     |

**Table 1.** Seeding occurrence for the model, with the seeding mass threshold indicated in brackets with units kg kg $^{-1}$ , and observations, with the ice crystal size used for the sublimation calculations marked within the brackets.

Figure 9. Full domain (over ocean and sea ice) mean occurrence frequency over time of MLC, SLC, and clear sky and the distribution of the number of cloud layers for MLCs with mass threshold of  $10^{-9}~{\rm kg\,kg^{-1}}$  to  $10^{-6}~{\rm kg\,kg^{-1}}$ .

almost exactly 10% compared to that at MOSAiC for each respective cloud-mass threshold. The SLC occurrence is between 23% and 32%, and the loss of MLCs with a higher cloud-mass threshold also increases the clear-sky fraction from 10% to 24%. We may further note that SLCs and CS cases increase at a similar rate with a higher cloud-mass threshold, indicating that the loss of MLCs either contributes to both classes or that, with a higher cloud-mass threshold, SLCs are classified as clear sky.

While highly sensitive to the thresholds and model assumptions used, a high frequency of MLCs that is comparable to the statistics of MOSAiC is found for the wider Arctic. The increased occurrence of CS might be the result of a less cloudy Arctic at lower latitudes. Separating the MLC fraction into the number of MLC layers in each profile, in the sub-categories of the MLC bar in Fig. 9, a high frequency of two-layered structures is found. Three-layered structures and MLC systems with more layers systematically decrease with a higher cloud-mass threshold.

At MOSAiC, similar distributions can be found (Fig. 7) where the whole modelled region with a cloud mass threshold of  $10^{-9} \text{ kg kg}^{-1}$ , likely coincidentally, is more similar to the observed frequencies using radiosonde classification alone. On the

other end, the observed MLCs (RS+Radar) at MOSAiC correspond to the full-domain MLC frequency using a cloud mass threshold of  $10^{-6} \text{ kg kg}^{-1}$ . The similarities in frequencies indicate a possibility of generalising the MLC occurrence to the wider Arctic region. A similar finding is presented in Achtert et al. (2025) based on their study of SLC and MLC occurrence across Arctic stations and research cruises. However, as advised in Achtert et al. (2025), care should be taken when using sporadic measurement data for drawing more general conclusions about cloud statistics over a larger area.

The spatial distributions of MLCs and SLCs during the simulated 32-day period are presented in Fig. 10. For each grid point, the mean occurrence frequency of MLCs and SLCs is calculated (including clear-sky samples). The MLC occurrence is substantial across most of the domain (> 50%) and shows no clear latitudinal dependency. An increase northward can be seen following the  $0^{\circ}$ E meridian. In contrast, following the  $180^{\circ}$ E meridian, no such trend can be seen. Neither do we see a clear correlation with the sea-ice extent shown in Fig. 1b. Rather, an increase in MLC can be seen away from coastlines. The SLC spatial occurrence in Fig. 10b is more evenly distributed, with a small increase in occurrence towards the coastlines and a decrease towards the high Arctic. The higher cloud mass thresholds show a similar, but lower frequency, pattern of MLCs, and the spatial distribution for the cloud mass threshold of  $10^{-6}$  kg kg $^{-1}$  is shown for completeness in Fig. B2.

Figure 10. Spatial occurrence of MLCs (a) and SLCs (b) over the Arctic domain for a cloud mass threshold of  $10^{-9} \text{ kg kg}^{-1}$ . Contours in (a) mark 60% (black) and 80% (grey) occurrence (of MLCs). In (b), the black contour marks a 20% occurrence (of SLCs).

## 6 Discussion and conclusions

Observed and modelled Arctic multilayer clouds are compared during autumn 2020 at the high-Arctic MOSAiC campaign site and across the Arctic region. Using a limited-area set-up in ICON, 32 days during August-September 2020 were simulated over a domain spanning 71°N - 90°N. A new immersion freezing parameterisation was developed to capture the regional autumnal

Arctic INPs. Overall, small impacts of the new INP parameterisation are seen. However, when isolating SLCs with cloud tops above 261K, a 14% increase in  $n_{\rm ice}$  is found. In general, we find that the model still encounters difficulties in the production of cloud ice at warmer temperatures of T > -12°C, though the reduction of heterogeneous nucleation at colder temperatures produces more realistic clouds in the upper atmosphere. We further tuned the CCN parameterisation (scaling by 0.01) to obtain comparable CCN concentrations in the model as measured in the Arctic. However, this introduces negligible differences to the original parameterisation. A discussion on the appropriateness of a saturation adjustment scheme when modelling Arctic clouds is beyond the scope of this study, but may be worth revisiting in the future.

Here, we want to address the questions raised in the introduction:

## 6.1 How does the ICON model with 2-moment microphysics perform in the high Arctic?

At the MOSAiC site, we find that the model adequately represents the observed occurrence rate of different cloud types. Mean temperature profiles are captured with a maximum mean difference of 0.8 K while moisture profiles are skewed towards too dry (moist) below (above) 2 km. A similar cold and dry bias in the lower atmosphere has been seen over Svalbard using ICON (Kiszler et al., 2023) and points to a more systematic problem. The model results show a tendency towards over-predicting the presence of high clouds and events of frozen precipitation. Boundary layer clouds are captured, but cloud ice mass content in lower clouds remains hard to capture with the model (Wallentin et al., 2025). Whether this is due to model limitations or uncertainty in retrievals at these warm  $(T > -5.5^{\circ}C)$  temperatures is difficult to disentangle. Modelled median LWP is about a factor of two less than observations. As the CCN scaling performed had no clear impacts on the cloud droplet number concentration, this underestimation may be due to other parameterised microphysical pathways. However, while statistical significance for the LWP difference is found between the model and the observations using the Mann-Whitney U test, the practical difference as indicated by the effect size is small. The frozen category median (including modelled cloud ice, graupel, and snow) is under-predicted by two orders of magnitude. However, some uncertainty must be taken into consideration as the retrieval of cloud ice at warmer temperatures is difficult. As the cloud ice across the domain only marginally increases with the new heterogeneous nucleation parameterisation, the lack of cloud ice is likely linked to other processes we do not yet understand. This is by far not a singular problem for ICON but has been seen in previous model studies (Morrison et al., 2009; Fridlind and Ackerman, 2017; Stevens et al., 2018) and the hypothesis calls for missing secondary ice production within the models (Sotiropoulou et al., 2024; Wallentin et al., 2025), especially at warmer temperatures.

# 6.2 What are the simulated MLC occurrence rates at the MOSAiC site and over the high Arctic?

We find that simulated multilayer clouds occur frequently in the region. The MLC occurrence across the domain and simulated period ranges from 44% to 67% with an occurrence frequency of SLCs varying between 32% and 23% as mass thresholds are varied from  $10^{-6} \,\mathrm{kg} \,\mathrm{kg}^{-1}$  to  $10^{-9} \,\mathrm{kg} \,\mathrm{kg}^{-1}$ . No clear latitudinal dependency is found. At the MOSAiC site, the observed MLC occurrence is found to be 46%-69%, depending on the inclusion of radar collocation (decrease in MLC occurrence). For the simulated MLC at the same site, an occurrence between 55% and 77% is found with values highly dependent on the

assumed cloud-mass threshold imposed in the model MLC algorithm. Due to limitations in both approaches when determining an MLC occurrence (see Achtert et al. (2025) for details on the observational findings), we may not say with certainty that the model is overestimating the occurrence for lower cloud-mass thresholds compared to the observations. Discrepancies may be due to differences in moisture, aerosol availability, and overall difficulties in representing MLCs in ICON (Wallentin et al., 2025). On the observational side, MLCs remain difficult to quantify due to the presence of seeding and instrument limitations. In general, two-layered systems are most common, followed by three-layered systems as previously found in satellite studies (Subrahmanyam and Kumar, 2017). Interestingly, the 2-layer occurrence is robust across mass thresholds and observational detection methods with a shared occurrence frequency of about 23%. Globally, two-layered systems occur with a 20% frequency (Subrahmanyam and Kumar, 2017), indicating that these cloud systems are easier to quantify across different methods.

We find large differences when putting our results into the context of earlier work. Another MOSAiC study by Silber and Shupe (2022) focuses on liquid-bearing MLCs and gives a 51% MLC occurrence that may serve as a lower bound of general (warm and mixed-phase) MLC occurrence. Achtert et al. (2025) applied the method of Vassel et al. (2019) to observations during MOSAiC, other research cruises to the central Arctic, and long-term land stations. MLC occurrence varied between 15% in October and 55% in March, while SLC occurrence varied between 20% in July and 70% in October. Within the MLC category, 30% to 70% of cases were seeding clouds. Findings on cloud occurrence from MOSAiC observations during August and September were also comparable to those from long-term observations at Ny-Ålesund, Svalbard (Achtert et al., 2025). During the MOCCHA campaign, a 54% occurrence was identified by Vüllers et al. (2021) using the same algorithm (while excluding warm (above 0°C) MLCs) and during a similar time of the year. Seasonal variations may thus play a large role in the variation of MLC occurrence. At lower latitudes, close to Svalbard, during the aircraft campaign PS106, an occurrence of 36% of MLCs was reported (Barrientos-Velasco et al., 2022) and at the research station at Ny Ålesund, Svalbard, frequencies of 29% (Vassel et al., 2019) and 44% (Nomokonova et al., 2019) were found. In contrast, in their investigation of cloudy profiles in the Arctic (60°N - 82°N) using satellite products, Liu et al. (2012) found an MLC frequency of only 20%. Satellite retrievals are, in general, uncertain for low clouds (Dietel et al., 2024) and may thus lead to a bias in the retrieved MLC occurrence over the Arctic (Schirmacher et al., 2023). These differences can partly be linked to variations in the definition of MLCs and calls for future efforts to harmonise the various methods used for MLC detection.

# 455 6.3 Do MLCs differ from SLCs? And if so, what are the major characteristics?

One of the major characteristics of MLCs is the occurrence of seeding. We find that seeding in the model occurs at a frequency of up to 46%. This is highly dependent on the seeding mass thresholds used in the developed algorithm for MLC detection. Seeding frequencies inferred from observational data (Vassel et al., 2019; Achtert et al., 2025) depend on the size of the ice crystals chosen for the sublimation calculation and vary between 59% and 27% for crystal diameters of 400 µm and 100 µm, respectively. Previous studies have found seeding frequencies of 23% over Svalbard (Vassel et al., 2019), 31% for cirrus clouds over the Swiss Alps (Proske et al., 2021), and 10% in a modelling study over the Swiss Alps (Dedekind et al., 2024). One may conclude that seeding remains a difficult mechanism to quantify. Comparing simulated and observed macrophysical charac-

teristics of MLCs, we find that MLCs tend to be thicker than SLCs. This is an indication that the microphysical interaction through seeding may be strengthening the clouds by increasing the cloud thickness. The degree of sublimation and, thus, seeding depends strongly on the fall distance between the layers (Achtert et al., 2025, Figure 8) and we find that the modelled and observed median gap thickness is about 700 m. The spread is small, only 8% of modelled clouds have gap thickness larger than 5 km, indicating that (only) cirrus overlaying stratocumulus clouds is rare. A 12% occurrence of thin layers (thickness 

485

# Appendix A: Statistical Evaluation

We perform the non-parametric Mann–Whitney U test (Mann and Whitney, 1947) for statistical significance and calculate the p-values and effect sizes to evaluate the differences between the model and the observational datasets. The effect sizes are calculated due to the large sample sizes of the categories, which may show statistical significance attributable to this size. The effect size calculated is the r value used for Mann–Whitney U tests (Tomczak and Tomczak, 2014) and the interpretation follows Pearson's correlation coefficient. r is calculated using the standardised Z-score:

$$\mu_U = \frac{n_1 n_2}{2} \tag{A1}$$

$$\sigma_U = \sqrt{\frac{n_1 n_2 (n_1 + n_2 + 1)}{12}} \tag{A2}$$

$$490 Z = \frac{U - \mu_U}{\sigma_U} (A3)$$

$$r = \frac{|Z|}{\sqrt{N}}, \quad \text{where} \quad N = n_1 + n_2$$
 (A4)

(A5)

where  $n_i$  is the size of the dataset i,  $\mu_U$  is the expected value of U under the null hypothesis, U is the U-statistic from the Mann-Whitney U test, and  $\sigma_U$  is the standard deviation.

Figure A1a shows the p-values for the comparison between the full (with time) datasets in the LWP and FWP categories. Statistical significance (p 

Figure A1. Mann-Whitney U test for statistical significance between the model and observations. Y-labels signify between which datasets the tests were performed, while x-labels show the variable in question. P-values are coloured such that statistically significant results ( $p \le 0.05$ ) are shaded in blue, and non-significant results (p > 0.05) are shaded in red. Boxes with values of 0.00 indicate, in general, very low ( $p 

# 500 Appendix B

Figure B1. Cloud ice mass concentration distribution for the H15 parameterisation and the *Arctic fit* filtered for cloud-top ice number concentrations of SLCs across the domain. Panel (a, warm) shows the distribution for warm temperatures (-12°C  $\leq$  T 

Figure B2. Spatial occurrence frequency of MLCs (a) and SLCs (b) over the Arctic domain for a cloud mass threshold of  $10^{-6} \text{ kg kg}^{-1}$ . Black contours signify 20% occurrence, grey contours 40%, and the white contour (only in (a)) 60% occurrence.

Author contributions. GW devised and ran the simulations, wrote the manuscript and led the analysis. LI and CH provided in-depth expertise. PA ran the observational MLC detection code for MOSAiC. All co-authors provided feedback on the manuscript. CH devised the overall project.

Competing interests. LI and MT are members of the Editorial Board of Atmospheric Chemistry and Physics.

Acknowledgements. The authors thank the Bundesministerium für Bildung und Forschung (BMBF) for funding the project with project numbers 03F0891A and 03F0891B. We further thank all those who contributed to MOSAiC and made this endeavour possible (Nixdorf et al., 2021). We acknowledge ACTRIS and the Finnish Meteorological Institute and thank them for providing observational data. This work was carried out on the HoreKa supercomputer funded by the Ministry of Science, Research and the Arts Baden-Württemberg and by the Federal Ministry of Education and Research. The authors would like to thank the Federal Ministry of Education and Research and the state governments (www.nhr-verein.de/unsere-partner) for supporting this project as part of the joint funding of National High-Performance Computing (NHR). This work was performed with the help of the Large Scale Data Facility at the Karlsruhe Institute of Technology, funded by the Ministry of Science, Research and the Arts Baden-Württemberg and by the Federal Ministry of Education and Research. GW is currently funded through the Horizon Europe project CleanCloud (Grant Agreement no. 101137639). LI is funded by the Chalmers Gender Initiative for Excellence (Genie). Grammarly and ChatGPT have been used for the stylistic and grammatical improvement of the manuscript.

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
