# Peer review of "The Prevalence of Arctic Multilayer Clouds and their Observed and Modelled Characteristics"

_EGUsphere, 2025_

## Referee Comment (RC1)

**Review of "The Prevalence of Arctic Multilayer Clouds and their Observed and Modelled Characteristics"**

by Wallentin et al.

**General comments:**

In this study, the authors analyze Arctic multilayer clouds (MLCs) using the ICON model and compare their occurrence, as well as their microphysical and macrophysical properties, with observations from the MOSAiC campaign. To account for the effects of locally emitted icenucleating particles, they additionally implemented and evaluated an immersion freezing parameterization within the model. The comparison between model results and observations indicates that ICON generally captures the occurrence of MLCs, although liquid and ice water paths are substantially underestimated.

The manuscript is logically structured and well written. I have some concerns regarding the comparison between the model and observations, particularly with respect to the occurrence of MLC and the occurrence of seeding. Nevertheless, this manuscript merits publication provided that the following comments are addressed.

**Specific comments:**

- My major concern with this study concerns the intercomparison of MLC occurrence and cloud properties between the model and the observations. The observationally based occurrence is derived from radiosondes (RS) and further supplemented by cloud radar observations (RS+Radar), whereas MLC detection in the model is based on cloud mass. The authors clearly demonstrate the sensitivity of MLC occurrence to the chosen cloud mass threshold. Ultimately, they select a threshold of 10-9 kg kg-1 for comparison with RS detections. However, the rationale for using RS-only detection rather than RS+Radar, which reduces the detection of spurious cloud layers, is unclear. Furthermore, it is not evident why the 10-9 kg kg-1 threshold was chosen, given that a threshold of 10-8 kg kg-1 appears to more closely match RS detections. Or is it because using a low threshold would ultimately lead to RS-like definition as only the saturation criterion is considered? In such a case, it would be important to have a consistent definition of saturation (see also bullet point 3). Similar concerns apply to the comparison of seeding occurrence (Tab. 1). One potential way to avoid the need to arbitrarily select a cloud mass threshold would be to employ a radar forward operator to generate radar reflectivity, enabling a more consistent comparison with the Achtert et al. (2025) detection algorithm.
- P7, L160: It took some time to realize that two distinct thresholds are used in this study: a cloud mass threshold and a seeding mass threshold, which share the same numerical values. It would be helpful to clearly distinguish these thresholds throughout the manuscript, for example, by using separate mathematical symbols, to avoid confusion. Additionally, it is unclear why the cloud mass threshold changes between sections (10-6 kg kg-1 in Section 5.1 vs. 10-9 kg kg-1 for MLC detection). While exploring the sensitivity of MLC properties to this threshold is valuable, once a threshold is chosen, it would be advisable to report all other microphysical properties

- (cloud droplet number concentration, ...) using the same threshold, unless there is a compelling reason not to do so.
- P7, L161-162: How do the authors decide with respect to which phase (liquid or ice) saturation is determined? Is this approach consistent with the method used to calculate saturation for cloud cover in ICON or with Achtert et al. (2025)?
- In this context, it would also be helpful to clarify whether the model employs a fractional or grid-scale cloud cover scheme, as this is not explicitly indicated in the model description.
- P9, L219-241: The reported values for mean cloud droplet number concentration (Nd) appear unusually high for the Arctic, even exceeding the number of observed cloud condensation nuclei that could potentially be activated (see Fig. 2). Could this be due to the mean being influenced by outliers, as seems to be the case for mean ice crystal number concentrations (see Fig. 3)? A similar concern applies to the reported cloud ice masses and number concentrations. I would suggest that reporting median values may provide a more robust representation of these quantities. Do the results differ when evaluating the medians instead of the means?
- P10, Fig. 3: In the caption, you state that values outside the interquartile range (IQR) are excluded, yet these values still appear to be included when calculating the means shown in the figure and subsequently reported in the manuscript. This also reinforces my earlier point: reporting median values would make the reported statistics less sensitive to outliers, potentially eliminating the need to filter out extreme values in the first place.
- P13, Fig. 5: As stated by the authors, liquid water content is not given in CloudNet if liquid-containing clouds have liquid-phased precipitation. I wonder how the median liquid water path has been derived for the model and for ShupeTurner. Were time steps with liquid-phase precipitation excluded from the comparison? If not, this may lead to a definition-inconsistent intercomparison, as the rainwater path is included in the model output and in Shupe—Turner, but not in CloudNet.
- P22, L463-464: Could you give more information about the physical pathway of this increase in geometrical cloud thickness?

**Minor Remarks:**

- P2, L39-41: While seeding can indeed initiate glaciation, neither riming nor secondary ice production can initiate it, since both processes require pre-existing cloud ice. I would therefore describe these processes as enhancing glaciation rather than initiating it. Similarly, the current phrasing suggests that the Wegener– Bergeron–Findeisen (WBF) process initiates glaciation, whereas it also primarily enhances glaciation once cloud ice is present. Consider rewording this part.
- P2, L50-51: Downwelling longwave radiation will only influence the lower cloud layer and not "each other".
- P3, L72-73: "... ICON Global analysis..." Are you referring to the analysis step (0th timestep) of the global forecast here? If so, I wonder whether this analysis is produced every 3 hours, as you further down state that you employ boundary conditions with 3-hourly updates.
- P3, L74-75: Here, one might understand that radiosondes are used as the only observations during the data assimilation. I assume you refer to the fact that the

radiosonde observations during MOSAiC are assimilated, in addition to the standard global observations. Furthermore, are observations really nudged (which I consider some kind of Newtonian relaxation) or simply used during the data assimilation step?

- P7, L165-166: No need to repeat the conditions, as you are referring to them in the first part of the sentence
- P8, L188-189: "... a standard deviation of the mean". Do you mean that the standard deviation is the same magnitude as the mean?
- P20, L413-414: Or because you are in an updraft-limited regime. On this end, I
  assume that grid-scale vertical velocity is used for aerosol activation, which might be
  too low at kilometer-scale resolution for Arctic clouds, which might be turbulencedriven.
- P21, L448: "... during the aircraft campaign PS106 ..." Isn't PS106 a ship cruise?

**References:**

Achtert, P., Seelig, T., Wallentin, G., Ickes, L., Shupe, M. D., Hoose, C., and Tesche, M.: Occurrence of seeding multi-layer clouds in the Arctic from ground-based observations, EGUsphere, https://doi.org/10.5194/egusphere-2025-3529, [preprint], 2025.

---

## Referee Comment (RC2)

The authors are performing an analysis of Multi-Layer Clouds (MLCs) in the Arctic using ICON model simulations over August 22 to September 23, 2020, and compare the model results to observations from the MOSAiC campaign. The authors use a newly developed *Arctic fit* immersion freezing parameterization, which includes an exponential for INP for colder temperatures and a second-degree polynomial fit for warmer temperatures. The model results show high sensitivity to mass thresholds in occurrences of MLCs and cloud seeding. The cloud occurrences and macrophysical quantities compare well with observations, even though the microphysical quantities are off often by orders of magnitude.

The manuscript is well written and presents novel work that contributes significantly to scientific progress. The manuscript is suitable for publication with minor revisions. The manuscript needs to address three aspects in detail.

1. The significant difference in Liquid Water Path and Frozen Water Path between the model and observations. Due to the high sensitivity on cloud mass thresholds, are the authors just coincidentally matching the observations on cloud occurrences, since the simulations have severe differences on quantities as fundamental as FWP and LWP?
2. The cloud seeding mechanism discussion could be expanded. The authors compare 1st layer MLCs and SLCs. A similar comparison could be made between seeded and non-seeded MLCs.
3. It would be helpful to include mean vertical profiles of clear-sky vs. SLCs vs. MLCs for more intercomparison. Also, differences in radiation between the SLC layers and MLCs could be reported, as that could be one of the main causes for the differences between 1st layer MLCs and SLCs.

Specific comments:

In figs. 3 & B1, why are the mean values outside the range? The values outside 1.5 times the IQR are excluded to simplify the interpretation, hence the mean should be within that range

The manuscript switches back and forth between using Celsius and Kelvin scale for temperatures. The readability could be improved by using a consistent unit (C?) for temperature throughout, and in cases where the other unit (K) needs to be used, provide the corresponding (C) values in parentheses.

Since the novelty of the work is focused on the newly developed *Arctic fit* immersion freezing parameterization, the authors should directly compare results from the ICON model using the Arctic fit parametrization and ICON model using the H15 (Hande et al. 2015) parametrization.

In Fig. 6, the temperature and qv are flipped in the figure compared to the caption and discussion.

Also, QV is referred to as specific humidity. QV is the water vapor mixing ratio, the ratio of water vapor mass to dry air mass, while specific humidity would be the ratio of water vapor to moist air mass.

In figs. 7 and 9, the authors should have 2-layer MLCs next to SLC, as SLCs are similar to 2-layer MLCs than >4 layer MLCs. I understand the authors want to focus on the robustness of the 2-layer MLCs across observations and models with different thresholds, but it's more sensible for the order to be CS, SLC, MLC (2), MLC (3), MLC (4), MLC (>4). The colors of CS and SLC could be switched for readability to have clear sky be represented by blue.

If the authors want to stress the robustness of 2-layer MLCs as more than just a coincidence, they need to do further analysis on what happens to the 3+ layer MLC regions when increasing the cloud mass threshold. What percentage of the 3+ layer MLC regions become CS / SLC / 2-layer MLCs upon increasing the threshold? What percentage of 2-layer MLC regions become CS / SLC upon increasing the threshold?

Line 291 "With a larger number of MLCs, either the SLCs or the clear-sky fraction has to decrease" - the lines before and after this mention the opposite, a decrease in MLCs and increase in SLC and clear-sky

One might argue that the "RS+Radar" observations are the best available data for comparison, as they are further validated by radar data. To that end, the authors should consider a cloud mass threshold of $10^{-5}$ kg/kg, which following the trends in threshold would be closer to the RS+Radar data

Lines 327-329 - "This may be explained by a 12% (25%) occurrence of modelled thin MLC (SLC) layers that are less than 100m thick. These cloud layers would not be included in the observational algorithm." - The authors should consider using the same cut-off thickness for the clouds as the observation algorithm.

Line 332 – "(39m and 22m difference in model and observations, respectively)" – It would be better to report these differences in median thicknesses as percentages.

Figs. 3, 5, 8 and B1 – Violin plots would give the readers a better understanding of the distribution of the microphysical and macrophysical quantities.

---

## Author Comment (AC1)

**Author Response to Reviewer #1**

The authors thank Reviewer #1 for their thorough review. We have addressed your comments. Please see the response to each point below in red. Figures in this reply are ordered with capital letters to distinguish them from the figures in the manuscript.

**Changes to the paper not discussed in the Authors' Comments:**

- MLCs above 0°C are treated differently in the observational algorithm and were not constrained by a 150m gap; this has now been rectified, and values have thus changed in Fig.7. Also, values for the RS product have changed due to an update in the observational algorithm. Please see Fig. A and B below.
- There were some inconsistencies in the development of the two algorithms; this has now been rectified. All model data has been updated with a 150m gap threshold. Overall, small changes are induced (MLC occurrence for 1E-9 kg/kg goes from 77% to 76%).
- Updated acknowledgements to follow the Supercomputer HoreKa suggested structure
- Updated Fig. A1 with [ ] brackets instead of ( ) for the units
- Wrong units in Fig. B1b

**Specific Comments**

- My major concern with this study concerns the intercomparison of MLC occurrence and cloud properties between the model and the observations. The observationally based occurrence is derived from radiosondes (RS and further supplemented by cloud radar observations (RS+Radar), whereas MLC detection in the model is based on cloud mass.

  The algorithm also asserts supersaturation with respect to ice or water (Line 161).

  The authors clearly demonstrate the sensitivity of MLC occurrence to the chosen cloud mass threshold. Ultimately, they select a threshold of 10-9 kg kg-1for comparison with RS detections. However, the rationale for using RS-only detection rather than RS+Radar, which reduces the detection of spurious cloud layers, is unclear. Furthermore, it is not evident why the 10-9 kg kg-1 threshold was chosen, given that a threshold of 10-8 kg kg-1 appears to more closely match RS detections.

  Yes, this choice may seem arbitrary. We chose 1E-9 kg/kg to follow the model's limit for radiation to interact with the clouds. However, we have now changed this following comments from Reviewer #2 and will consider the cloud mass threshold of 1E-5 kg/kg mainly, but also showing medians for cloud height, thickness, and gaps for the other thresholds to remove the arbitrary part of the comparison.

We further add the RS+Radar cloud heights and thicknesses for the model comparison instead of the RS only, and include all cloud mass thresholds (CMTs) to remove this absolute comparison to a certain CMT.

[Figure]

Fig A (Fig.7) with new colours, added CMT5 and updates to the observational product.

[Figure]

Fig. B (Fig.8) Cloud thickness and cloud gap thresholds are more consistently treated in the RS+Radar product. CT heights are shifted up due to the removal of cloud layers below the lowest radar gate (see Achtert et al. (2025) for details). Outlines of the distributions are added on request from Reviewer #2. A filtered model data category has been added, where clouds with a thickness of less than 100m are excluded.

Or is it because using a low threshold would ultimately lead to RS-like definition as only the saturation criterion is considered? In such a case, it would be important to have a consistent definition of saturation (see also bullet point 3).

Please see the answer below regarding the saturation definition.

Similar concerns apply to the comparison of seeding occurrence (Tab. 1).

We have now also added the seeding occurrence for all cloud mass thresholds to be less deterministic in terms of choosing a "correct" cloud mass threshold

One potential way to avoid the need to arbitrarily select a cloud mass threshold would be to employ a radar forward operator to generate radar reflectivity, enabling a more consistent comparison with the Achtert et al. (2025t detection algorithm.

Yes, this would have been a good idea to implement. Unfortunately, we do not have all the output parameters required to employ a radar forward operator at this stage, and new simulations cannot be performed due to the computational constraints. We hope that the further comparison with threshold 1E-5 kg/kg and adding the uncertainty (in terms of all mass thresholds) in the following analysis addresses the reviewer's concerns.

- **P7, L160**: It took some time to realize that two distinct thresholds are used in this study: a cloud mass threshold and a seeding mass threshold, which share the same numerical values. It would be helpful to clearly distinguish these thresholds throughout the manuscript, for example, by using separate mathematical symbols, to avoid confusion.

  Yes, this is a fair point. We have updated the manuscript to refer to a cloud mass threshold (CMT) and a seeding mass threshold (SMT) throughout the text.

  Additionally, it is unclear why the cloud mass threshold changes between sections (10-6 kg kg-1 in Section 5.1 vs. 10 -9 kg kg-1 for MLC detection). While exploring the sensitivity of MLC properties to this threshold is valuable, once a threshold is chosen, it would be advisable to report all other microphysical properties (cloud droplet number concentration, …) using the same threshold, unless there is a compelling reason not to do so.

  We used our previous greatest cloud mass threshold to better ascertain whether any changes were seen within the clouds with the new parameterisations. We generally agree on consistency; however, here we are evaluating whether there is a general response to the change in parameterisations rather than for a cloud mass threshold that is tuned to observations. We can, however, give a range of values. Thus, we evaluate CMT5 and CMT9 for the two parameterisations and plot these in Fig. C. We find a 2%-16% difference in cloud ice at warm temperatures with CMT5 and CMT9, respectively. We update the section accordingly.

[Figure]

Fig. C (Fig. 3) The boxplots show the distribution of CMT5 and mean (diamonds) and median (dashed line) for CMT9. Violin plots (on request from Reviewer #2) show the distribution of CMT5.

**P7, L161–162:** How do the authors decide with respect to which phase (liquid or ice) saturation is determined? Is this approach consistent with the method used to calculate saturation for cloud cover in ICON or with Achtert et al. (2025)?

For both algorithms, both saturations are taken into account. The clause is implemented with an OR statement such that when saturation for either liquid or ice is reached, the layer is flagged as a cloud layer. In the model, this is then combined with a cloud mass threshold (LWC+IWC) such that we ensure cloud mass is present. This allows for both fully liquid, fully ice, and mixed-phase clouds to be flagged. In the observational algorithm, the presence of clouds is validated using radar. The model approach is similar to Achtert et al. (2025), but the measured variables come with uncertainty that we do not take into account here. We have not aligned the algorithm with the cloud cover scheme in ICON.

In this context, it would also be helpful to clarify whether the model employs a fractional or grid-scale cloud cover scheme, as this is not explicitly indicated in the model description.

For the radiation calculation, we use a diagnostic cloud cover, a type of fractional cloud cover scheme. In the microphysics scheme, a simple grid-scale approach is used (0 or 1).

- **P9, L219–241:** The reported values for mean cloud droplet number concentration (Ndt appear unusually high for the Arctic, even exceeding the number of observed cloud condensation nuclei that could potentially be activated (see Fig. 2). Could this

be due to the mean being influenced by outliers, as seems to be the case for mean ice crystal number concentrations (see Fig. 3)?

Yes, thank you for catching this. There was a bug in the calculation for the units. We update the section with a table showcasing the mean and median number concentrations for cloud liquid and ice. The largest mean is 40 cm^-3, which is more in line with what the parameterisation allows for.

A similar concern applies to the reported cloud ice masses and number concentrations. I would suggest that reporting median values may provide a more robust representation of these quantities. Do the results differ when evaluating medians instead of means?

The medians are indicated in Fig. 3 by the horizontal lines, and as stated in Line 231 these medians remain larger for H15 than for the Arctic fit. We only find larger values in the mean values due to these outliers, as you also mention. Thus, as we try to state, the impact of the parameterisation is limited.

We also update this section with the CMT5-CMT9 threshold and the respective new values in the table mentioned above. The new Figure 3 is shown above in Fig. C.

- **P10, Fig. 3:** In the caption, you state that values outside the interquartile range (IQR) are excluded, yet these values still appear to be included when calculating the means shown in the figure and subsequently reported in the manuscript. This reinforces the earlier concern: using medians would make the reported statistics less sensitive to outliers and might remove the need to filter extreme values in the first place.

  A slight miswording, the outliers are only visually excluded to aid the interpretation of the figure. We add to each figure with boxplots:

  **"Values larger (or smaller) than these are outliers and are not shown (only in the figure) to simplify the visual interpretation."**

- **P13, Fig. 5:** As stated by the authors, liquid water content is not given in CloudNet if liquid-containing clouds have liquid-phase precipitation. I wonder how the median liquid water path has been derived for the model and for Shupe–Turner. Were time steps with liquid-phase precipitation excluded from the comparison? If not, this may lead to a definition-inconsistent intercomparison, as the rainwater path is included in the model output and in Shupe–Turner, but not in CloudNet.

  The ShupeTurner algorithm employs a similar method as they're bound by the same observations. We add to methods:

  **"LWC is not available during liquid-phase precipitation in both of these retrievals."**

We also update the LWP statement as we are currently using the integrated LWC and not the measured LWP.

Line 187: **"Liquid water path (LWP) and ice water path (IWP) are calculated as the column-integrated LWC and IWC, respectively. The uncertainty in LWC is 15% to 25%. "**

The comparison to observations is always difficult. We included the rainwater path in the modelled LWP, as this is rain within the clouds and not precipitation reaching the surface. For simplicity, we may remove it. The median is marginally reduced by about 6% and the factor differences remain the same.

[Figure]

Fig D, (Fig. 5), where LWP now only contains liquid water and no in-cloud rain.

**P22, L463–464:** Could you provide more information about the physical pathway responsible for the increase in geometrical cloud thickness?
To be clear, there is no "increase" in the thickness; rather, we find that MLCs are thicker than SLCs. The 'strengthening' we are hypothesising refers to this finding. For simplicity and to refrain from making any hypotheses in the manuscript, this sentence is removed.

**Minor Remarks**

- **P2, L39–41:** While seeding can indeed initiate glaciation, neither riming nor secondary ice production can initiate it, since both processes require pre-existing cloud ice. These processes should therefore be described as enhancing glaciation rather than initiating it. Similarly, the current phrasing suggests that the Wegener–Bergeron–Findeisen (WBF) process initiates glaciation, whereas it primarily enhances glaciation once cloud ice is present. Consider rewording this section.

  Perhaps the word "initiate" is a bit misleading here. We have changed this to:

**"Glaciation, the transition from a mixed-phase state to fully ice, may be enhanced by the seeding of frozen precipitation (ice crystals, snow, or graupel) together with riming and secondary ice production (SIP), through the Wegener-Bergeron-Findeisen (WBF) mechanism"**

- **P2, L50–51:** Downwelling longwave radiation will only influence the lower cloud layer and not "each other."

  Changed to **"Radiatively, overlaying cloud layers in MLC systems influence lower clouds through an increase in downwelling longwave radiation."**

- **P3, L72–73:** "ICON Global analysis": Are you referring to the analysis step (0th timestep) of the global forecast here? If so, I wonder whether this analysis is produced every 3 hours, as you further down state that you employ boundary conditions with 3-hourly updates.

  Yes, we initialise the model at 00UTC (Line 87) from the analysis. We rewrite Line 74 to make the product clearer:

  **"The ICON Global analysis is a combination of forecast and data assimilation. Every 3 hours, a new data-assimilation cycle is initiated using the global observing network and local data assimilation from the radiosoundings during MOSAiC. Thus, we maintain a close agreement to observations at initialisation and boundary conditions supply changes along the domain edge with 3-hourly updates. "**

- **P3, L74–75:** Here, one might understand that radiosondes are used as the only observations during the data assimilation. I assume you refer to the fact that the radiosonde observations during MOSAiC are assimilated, in addition to the standard global observations. Furthermore, are observations really nudged (which I consider some kind of Newtonian relaxation) or simply used during the data assimilation step?

  No, they're simply used in the data assimilation. Thank you for catching this misuse of the word. The rewrite is listed above, P3, L72-73.

- **P7, L165–166:** No need to repeat the conditions, as you are already referring to them in the first part of the sentence.

  Ok, removed.

- **P8, L188–189:** "A standard deviation of the mean." Do you mean that the standard deviation is the same magnitude as the mean?
  We excuse the ambiguity, this has now been removed to also better reflect the fact that we are currently using the integrated LWC and not the measured LWP.

Line 187: **"Liquid water path (LWP) and ice water path (IWP) are calculated as the column-integrated LWC and IWC, respectively. The uncertainty in LWC is 15% to 25%. "**

- **P20, L413–414:** Or because you are in an updraft-limited regime. On this end, I assume that grid-scale vertical velocity is used for aerosol activation, which might be too low at kilometer-scale resolution for Arctic clouds, which might be turbulence-driven.

  That is a very good point. We add: **"It may also be due to too low vertical velocities, limiting the cloud droplet activation."**

- **P21, L448:** "... during the aircraft campaign PS106 ..." Isn't PS106 a ship cruise?

  Yep, you're very correct. Thanks for catching this. We update:

  **"At lower latitudes, close to Svalbard, during the PS106 campaign, an occurrence of 36% of MLCs was reported…"**

---

## Author Comment (AC2)

**Author Response to Reviewer #2**

The authors thank Reviewer #2 for their detailed review. We have addressed your comments. Please see the response to each point below in red. Figures in this reply are ordered with capital letters to distinguish them from the figures in the manuscript.

**Changes to the paper not discussed in the Authors' Comments:**
-   We found an inconsistency in the way the warm (above 0°C) clouds are treated in the RS+Radar product. These clouds were not constrained by a 150m gap. The same threshold as for cold clouds (>150m gap) is now applied, and thus, RS+Radar is updated accordingly in Fig. B. Also, values for the RS product have changed due to an update in the observational algorithm.
-   There were some inconsistencies in the development of the two algorithms; this has now been rectified. All model data has been updated with a 150m gap threshold. Overall, small changes are induced (MLC occurrence for 1E-9 kg/kg goes from 77% to 76%).
-   Updated acknowledgements to follow the HoreKa suggested structure
-   Updated Fig. A1 with [ ] brackets instead of ( ) for the units
* * *
**Major Points**

1.   The significant difference in Liquid Water Path and Frozen Water Path between the model and observations. Due to the high sensitivity on cloud mass thresholds, are the authors just coincidentally matching the observations on cloud occurrences, since the simulations have severe differences on quantities as fundamental as FWP and LWP?

We can acknowledge the significant differences in FWP and LWP and understand the point made by the reviewer. Modelling Arctic clouds is a challenging topic, and many models (Line 420) struggle with capturing these fundamental variables.
As can be seen in Fig. 4, the overall structure does compare quite well between the model and the observations. Thus, we continue with the assumption that the model is adequately capturing the clouds as seen in the observations. The LWP/FWP bias is therefore rather seen as an artefact of "too weak" clouds rather than a completely different set of clouds.

However, as the comment might also refer to the comparison of modelled clouds with a cloud mass threshold of 1E-9 kg/kg to the observed clouds, we have expanded on this section and are now including a higher cloud mass threshold (now referred to as CMT) of 1E-5 kg/kg, as the reviewer suggested below, that better corresponds to the observed MLC frequency, Fig.7 and Fig. 8 are thus updated accordingly. The following analysis and the seeding table are also extended to show all CMTs in an effort to be less absolute in terms of choosing a CMT. See figures B and D below.

2. The cloud seeding mechanism discussion could be expanded. The authors compare 1st layer MLCs and SLCs. A similar comparison could be made between seeded and non-seeded MLCs.

Yes, this is an interesting aspect of these clouds. We have another paper in preparation that focuses on the differences between seeded and non-seeded MLCs and SLCs, where we explore this from a model perspective. As observations only indicate a possibility of seeding and not deterministically, we choose to omit the observations for further analysis in regards to seeded/non-seeded clouds.

3. It would be helpful to include mean vertical profiles of clear-sky vs. SLCs vs. MLCs for more intercomparison.

We continue with the assumption that this comment pertains to a temperature profile, as no other variable was specified. Below in Fig. A are the modelled mean vertical profiles of temperature for CS, SLC, and MLC profiles. Differences are only distinguishable in the lowest ~3km. A clear temperature inversion can be seen between 500 m and 1000 m in the CS profile. A gradual weakening of the stability is seen in the mean profile for SLCs and MLCs. 10m temperature (lowest model level) differs between -6°C for CS, -3°C for SLCs, and -2°C for MLCs. We currently do not see where this evaluation would fit into the manuscript and will keep this analysis in the Authors' comments.

[Figure]

Fig. A Mean vertical temperature profiles.

Also, differences in radiation between the SLC layers and MLCs could be reported, as that could be one of the main causes for the differences between 1st layer MLCs and SLCs.

This aspect is also an interesting one when it comes to MLCs. In the paper in preparation, mentioned above, we also look into the radiative interaction between MLC/SLC and seeded/non-seeded clouds from a model perspective. This is also difficult to obtain from a ground-based observational perspective and is thus not further investigated here.

Specific comments:

In figs. 3 & B1, why are the mean values outside the range? The values outside 1.5 times the IQR are excluded to simplify the interpretation, hence the mean should be within that range

A slight miswording, the outliers are only visually excluded to aid the interpretation of the figure. Thank you for catching this.
We add to each figure with boxplots:
**"Values larger (or smaller) than these are outliers and are excluded (only in the figure) to simplify the visual interpretation."**

The manuscript switches back and forth between using Celsius and Kelvin scale for temperatures. The readability could be improved by using a consistent unit (C?) for temperature throughout, and in cases where the other unit (K) needs to be used, provide the corresponding (C) values in parentheses.

Yes, thank you for noticing. Due to the fact that the parameterisation is in Kelvin, we will keep the units for this part. However, we add the corresponding values in Celsius throughout section 2.1.1, 5.1, and Fig.6

Since the novelty of the work is focused on the newly developed Arctic fit immersion freezing parameterization, the authors should directly compare results from the ICON model using the Arctic fit parametrization and ICON model using the H15 (Hande et al. 2015) parametrization.

This was done in Section 5.1

In Fig. 6, the temperature and qv are flipped in the figure compared to the caption and Discussion.

Thank you for noticing. The figure and captions are now matching.

Also, QV is referred to as specific humidity. QV is the water vapor mixing ratio, the ratio of water vapor mass to dry air mass, while specific humidity would be the ratio of water vapor to moist air mass.

Thank you for noticing. Fixed.

In figs. 7 and 9, the authors should have 2-layer MLCs next to SLC, as SLCs are similar to 2-layer MLCs than >4 layer MLCs. I understand the authors want to focus on the robustness of the 2-layer MLCs across observations and models with different thresholds, but it's more sensible for the order to be CS, SLC, MLC (2), MLC (3), MLC (4), MLC (>4). The colors of CS and SLC could be switched for readability to have clear sky be represented by blue.

We do not agree with the justification for swapping the order. The main part of the figure is reading the MLC occurrence, which gets increasingly problematic by swapping these to the top of the plot. Furthermore, the SLC frequency is similar to 2-layered clouds, but we believe this does not justify a less readable plot for the values that are discussed in the text.

The colour change is a good suggestion, and CS is now in blue. Following the reviewer comments below, we have added a new mass threshold, 1E-5 kg/kg. Now these cloud mass thresholds are referred to as CMT.

[Figure]

Fig B (Fig.7) with new colours and added CMT5 with updated values for the observational algorithm.

If the authors want to stress the robustness of 2-layer MLCs as more than just a coincidence, they need to do further analysis on what happens to the 3+ layer MLC regions when increasing the cloud mass threshold. What percentage of the 3+ layer MLC regions become CS / SLC / 2-layer MLCs upon increasing the threshold? What percentage of 2-layer MLC regions become CS / SLC upon increasing the threshold?

This is an interesting question. We have extended our analysis to look at this. We find a gradual change of cloud layers, most easily visualised in Fig. C below. For each threshold transition, CMT5 to CMT6, CMT6 - CMT7, etc, we have calculated the number of profiles (during the radiosonde times) that change cloud layer number or remain in the same category (CS, SLC, MLC2, MLC3, etc). We find that a majority of timesteps remain in the same category; meanwhile, the transition to a greater cloud number happens between 5-30% of the profiles. For a small number of occurrences, the cloud number is reduced with a lower CMT. This happens when two layers are vertically close with a relatively high cloud mass between the layers; thus, when decreasing the CMT, these two layers merge into one. This happens more frequently for categories with a greater cloud number. In general, the dominant category has more cloud layers that remain within that category. No clear trends when looking at the MLC2 category can be discerned. We thus refrain from making any hypotheses on whether this occurrence is purely random, constrained by the algorithms used to classify them, or a physical phenomenon.

We edit the manuscript accordingly.
**"... indicating that these cloud systems are easier to quantify across different methods"** is removed
And we change:
"**The similarities in the representation of the approximately 22% occurrence of**

**two-layered clouds may be purely coincidental, constrained by the classifying algorithms, or a physical phenomenon we do not yet understand."**

[Figure]

Fig C. Number of transitions for each CMT to the next lower CMT for CS, SLC, and the first three MLC categories. The colour indicates how many timesteps, identified with a higher CMT, are in the same or a different category for a lower CMT.

Line 291 "With a larger number of MLCs, either the SLCs or the clear-sky fraction has to decrease" - the lines before and after this mention the opposite, a decrease in MLCs and increase in SLC and clear-sky

Yes, we've inverted the sentence: **"With a smaller number of MLCs, either the SLCs or the clear-sky fraction has to increase."**

One might argue that the "RS+Radar" observations are the best available data for comparison, as they are further validated by radar data. To that end, the authors should consider a cloud mass threshold of 10^-5 kg/kg, which following the trends in threshold would be closer to the RS+Radar data

We add a new CMT, CMT5 (1E-5 kg/kg) and have now updated Fig. 7 (See Fig. B above) and the following analysis on the heights and thicknesses of the clouds based on this larger threshold. We furthermore add the other CMTs to Fig. 8 (Fig. D below), and we report seeding occurrence for all CMTs.

Lines 327-329 - "This may be explained by a 12% (25%) occurrence of modelled thin MLC (SLC) layers that are less than 100m thick. These cloud layers would not be included in the observational algorithm." - The authors should consider using the same cut-off thickness for the clouds as the observation algorithm.

For the cloud thickness, a 100m thickness for the observational algorithm was imposed to remove layers that may be due to uncertainty in the instruments. For the model, this uncertainty does not apply, and thus, we have neglected this threshold on the model data for the MLC detection, as it has no physical meaning. Furthermore, due to the discrete model levels that increase with height, the majority of the filtered clouds would be in the boundary layer. To show this, we filter the model data for thicknesses below 100m, the plot is shown in Fig. D below.
Overall, small shifts can be seen in the model data, but it is a more accurate comparison, so we update the manuscript accordingly.

[Figure]

Fig. D (Fig.8) Cloud thickness and cloud gap thresholds are more consistently treated in the RS+Radar product. CT heights are shifted up due to the removal of cloud layers below the lowest radar gate (see Achtert et al. (2025) for details.)

The sentence in question is updated for the CMT5-CMT9 comparison.
**"For the unfiltered data, this may be explained by a 12% - 32% (25%-31%) occurrence of modelled thin MLC (SLC) layers that are less than 100m thick for CMT5 - CMT9. For the filtered data, discrepancies may be due to the significant differences in water paths discussed above."**

We also added a clarification.
**"Due to uncertainty, cloud layers below 100m are not included in the observational algorithm. We thus filter the clouds identified in the model algorithm similarly. Figure 9a shows both the unfiltered and the filtered model data ("Model 100m")."**

Line 332 – "(39m and 22m difference in model and observations, respectively)" – It would be better to report these differences in median thicknesses as percentages.

Ok. Updated the sentence also to include the CMTs and the filtered model data (Model 100m).
**"However, we find that the median thickness of the 1st Layer MLC is similar to SLCs for the model (SLCs are 7% smaller - 0.8% larger (CMT5 - CMT9)). For the observations, the lowest layer is 33% thinner than SLCs."**

Figs. 3, 5, 8 and B1 – Violin plots would give the readers a better understanding of the distribution of the microphysical and macrophysical quantities.

As we mainly discuss the medians in these figures, we'd like to keep the boxplots for readability, but we've added a violin plot outline in the background of each boxplot. See plots below (and Fig. D above).

[Figure]

Fig. E (Fig. 3) with updates; CMT5 is shown in the boxplot and with the distribution in the violin plot. CMT9 means (diamonds) and medians (dashed black lines) are further shown for comparison.

[Figure]

Fig. F (Fig. 5)

[Figure]

Fig. G (Fig. B1). With the same updates as Fig. D and with rectified units in b.